# Visualizing anatomically registered data with brainrender

Federico Claudi[1]*, Adam L Tyson[1], Luigi Petrucco[2,3], Troy W Margrie[1], Ruben Portugues[2,3,4], Tiago Branco[1]*

[1]UCL Sainsbury Wellcome Centre, London, United Kingdom; [2]Institute of Neuroscience, Technical University of Munich, Munich, Germany; [3]Max Planck Institute of Neurobiology, Research Group of Sensorimotor Control, Martinsried, Germany; [4]Munich Cluster for Systems Neurology (SyNergy), Munich, Germany

**Abstract** Three-dimensional (3D) digital brain atlases and high-throughput brain-wide imaging techniques generate large multidimensional datasets that can be registered to a common reference frame. Generating insights from such datasets depends critically on visualization and interactive data exploration, but this a challenging task. Currently available software is dedicated to single atlases, model species or data types, and generating 3D renderings that merge anatomically registered data from diverse sources requires extensive development and programming skills. Here, we present brainrender: an open-source Python package for interactive visualization of multidimensional datasets registered to brain atlases. Brainrender facilitates the creation of complex renderings with different data types in the same visualization and enables seamless use of different atlas sources. High-quality visualizations can be used interactively and exported as high-resolution figures and animated videos. By facilitating the visualization of anatomically registered data, brainrender should accelerate the analysis, interpretation, and dissemination of brain-wide multidimensional data.

**\*For correspondence:**
federico.claudi.17@ucl.ac.uk (FC);
t.branco@ucl.ac.uk (TB)

**Competing interests:** The authors declare that no competing interests exist.

## Introduction

Understanding how nervous systems generate behavior benefits from gathering multidimensional data from different individual animals. These data range from neural activity recordings and anatomical connectivity, to cellular and subcellular information such as morphology and gene expression profiles. These different types of data should ideally all be in register so that, for example, neural activity in one brain region can be interpreted in light of the connectivity of that region or the cell types it contains. Such registration, however, is challenging. Often it is not technically feasible to obtain multidimensional data in a single experiment, and registration to a common reference frame must be performed post hoc. Even for the same experiment type, registration is necessary to allow comparisons across individual animals (*Simmons and Swanson, 2009*).

While different types of references can in principle be used, neuroanatomical location is a natural and most commonly used reference frame (*Chon et al., 2019*; *Oh et al., 2014*; *Arganda-Carreras et al., 2018*; *Kunst et al., 2019*). In recent years, several high-resolution three-dimensional (3D) digital brain atlases have been generated for model species commonly used in neuroscience (*Wang et al., 2020*; *Oh et al., 2014*; *Arganda-Carreras et al., 2018*; *Kunst et al., 2019*). These atlases provide a framework for registering different types of data across macro- and microscopic scales. A key output of this process is the visualization of all datasets in register. Given the intrinsically 3D geometry of brain structures and individual neurons, 3D renderings are more readily understandable and can provide more information when compared to two dimensional images. Exploring interactive 3D visualizations of the brain gives an overview of the relationship between datasets and brain regions and helps generating intuitive insights about these relationships. This is

particularly important for large-scale datasets such as the ones generated by open-science projects like MouseLight (*Winnubst et al., 2019*) and the Allen Mouse Connectome (*Oh et al., 2014*). In addition, high-quality 3D visualizations facilitate the communication of experimental results registered to brain anatomy.

Generating custom 3D visualizations of atlas data requires programmatic access to the atlas. While some of the recently developed atlases provide an API (Application Programming Interface) for accessing atlas data (*Wang et al., 2020*; *Kunst et al., 2019*), rendering these data in 3D remains a demanding and time-consuming task that requires significant programming skills. Moreover, visualization of user-generated data registered onto the atlas requires an interface between the user data and the atlas data, which further requires advanced programming knowledge and extensive development. There is therefore the need for software that can simplify the process of visualizing 3D anatomical data from available atlases and from new experimental datasets.

Currently, existing software packages such as cocoframer (*Lein et al., 2007*), BrainMesh (*Yaoyao, 2020*), and SHARPTRACK (*Shamash et al., 2018*) provide some functionality for 3D rendering of anatomical data. These packages, however, are only compatible with a single atlas and cannot be used to render data from different atlases or different animal species. Achieving this requires adapting the existing software to the different atlases datasets or developing new dedicated software all together, at the cost of significant additional efforts, often duplicated. An important limitation of the currently available software is that it frequently does not support rendering of non-atlas data, such as data from publicly available datasets (e.g. MouseLight) or produced by individual laboratories. This capability is essential for easily mapping newly generated data onto brain anatomy at high resolution and produce visualizations of multidimensional datasets. More advanced software such as natverse (*Bates et al., 2020*) offers extensive data visualization and analysis functionality, but currently, it is mostly restricted to data obtained from the *Drosophila* brain. Simple Neurite Tracer (*Arshadi et al., 2020*), an ImageJ-based software, can render neuronal morphological data from public and user-generated datasets and is compatible with several reference atlases. However, this software does not support visualization of data other than neuronal morphological reconstructions nor can it be easily adapted to work with different or new atlases beyond the ones already supported. Finally, software such as MagellanMapper (*Young et al., 2020*) can be used to visualize and analyze large 3D brain imaging datasets, but the visualization is restricted to one data item (i.e. images from one individual brain). It is therefore not possible to combine data from different sources into a single visualization. Ideally, a rendering software should work with 3D mesh data instead of 3D voxel image data to allow the creation of high-quality renderings and facilitate the integration of data from different sources.

An additional consideration is that existing software tools for programmatic neuroanatomical renderings have been developed in programming languages such as R and Matlab, and there is currently no available alternative in Python. The popularity of Python within the neuroscientific community has grown tremendously in recent years (*Muller et al., 2015*). Building on Python's simple syntax and free, high-quality data processing and analysis packages, several open-source tools directly aimed at neuroscientists have been written in Python and are increasingly used (e.g., *Mathis et al., 2018*; *Pachitariu et al., 2017*; *Tyson and Rousseau, 2020b*). Developing a python-based software for universal generation of 3D renderings of anatomically registered data can therefore take advantage of the increasing strength and depth of the python neuroscience community for testing and further development.

For these reasons, we have developed brainrender: an open-source python package for creating high-resolution, interactive 3D renderings of anatomically registered data. Brainrender is written in Python and integrated with BrainGlobe's AtlasAPI (*Claudi et al., 2020*) to interface natively with different atlases without need for modification. Brainrender supports the visualization of data acquired with different techniques and at different scales. Data from multiple sources can be combined in a single rendering to produce rich and informative visualizations of multidimensional data. Brainrender can also be used to create high-resolution, publication-ready images and videos (see *Tyson and Rousseau, 2020b*; *Adkins et al., 2020*), as well as interactive online visualizations to facilitate the dissemination of anatomically registered data. Finally, using brainrender requires minimal programming skills, which should accelerate the adoption of this new software by the research community. All brainrender code is available at the GitHub repository together with extensive online documentation and examples.

## Results

### Design principles and implementation

A core design goal for brainrender was to generate a visualization software compatible with any reference atlas, thus providing a generic and flexible tool (*Figure 1A*). To achieve this goal, brainrender has been developed as part of the BrainGlobe's computational neuroanatomy software suite. In particular, we integrated brainrender directly with BrainGlobe's AtlasAPI (*Claudi et al., 2020*). The AtlasAPI can download and access atlas data from several supported atlases in an unified format. Brainrender uses the AtlasAPI to access 3D mesh data from individual brain regions as well as metadata about the hierarchical organization of the brain's structures (*Figure 1B*). Thus, the same programming interface can be used to access data from any atlas (see code examples in *Figure 2*), including recently developed ones (e.g. the enhanced and unified mouse brain atlas, *Chon et al., 2019*).

The second major design principle was to enable rendering of any data type that can be registered to a reference atlas, either from publicly available datasets or from individual laboratories. Brainrender can directly visualize data produced with any analysis software from the BrainGlobe suite, including cellfinder (*Tyson et al., 2020a*) and brainreg (*Tyson and Rousseau, 2020b*). In addition, brainrender provides functionality for easily loading and visualizing commonly used data types such as .npy files with cell coordinates or image data, .obj, and .stl files with 3D mesh data and .json files with streamlines data for mesoscale connectomics. Additional information about the file formats accepted by brainrender can be found in the online documentation. Brainglobe's software suite also includes imio which can load data from several file types (e.g. tiff and .nii), and additional file formats can be loaded through the numerous packages provided by the python ecosystem. Finally, the existing loading functionality can be easily expanded to support user-specific needs by directly plugging in custom user code into the brainrender interface (*Figure 1A*).

One of the goals of brainrender is to facilitate the creation of high-resolution images, animated videos, and interactive online visualizations from any anatomically registered data. Brainrender uses vedo as the rendering engine (*Musy et al., 2019*), a state-of-the-art tool that enables fast, high-quality rendering with minimal hardware requirements.

High-resolution renderings of rich 3D scenes can be produced rapidly (e.g. 10,000 cells in less than 2 s) in standard laptop or desktop configurations. Benchmarking tests across different operating systems and machine configurations show that using a GPU can increase the framerate of interactive renderings by a factor of 3.5 (see *Tables 1* and *2* in Materials and methods). This performance increase, however, depends on the complexity of the pre-processing steps, such as data loading and mesh generation, which run on the CPU. As one the main goals of brainrender is to produce high-resolution visualizations, we have made the rendering quality independent of hardware configuration, which only affects the rendering time. Animated videos and online visualizations can be produced with a few lines of code in brainrender. Several options are provided for easily customizing the appearance of rendered objects, thus enabling high-quality, rich data visualizations that combine multiple data sources.

Finally, we aimed for brainrender to empower scientists with little or no programming experience to generate advanced visualizations of their anatomically registered data. To make brainrender as user-friendly as possible we have produced extensive documentation, tutorials and examples for installing and using the software. We have also developed a graphic user interface (GUI) to access most of brainrender's core functionality. This GUI can be used to perform actions such as rendering of brain regions and labeled cells (e.g. from cellfinder) and creating images of the rendered data, without writing custom python code (*Figure 1C*), (*Video 1*).

### Visualizing brain regions and other structures

A key element of any neuroanatomical visualization is the rendering of the entire outline of the brain as well as the borders of brain regions of interest. In brainrender, this can easily be achieved by specifying which brain regions to include in the rendering. The software will then use BrainGlobe's AtlasAPI to load the 3D data and subsequently renders them (*Figure 1B*).

Brainrender can also render brain areas defined by factors other than anatomical location, such as gene expression levels or functional properties. These can be loaded either directly as 3D mesh data

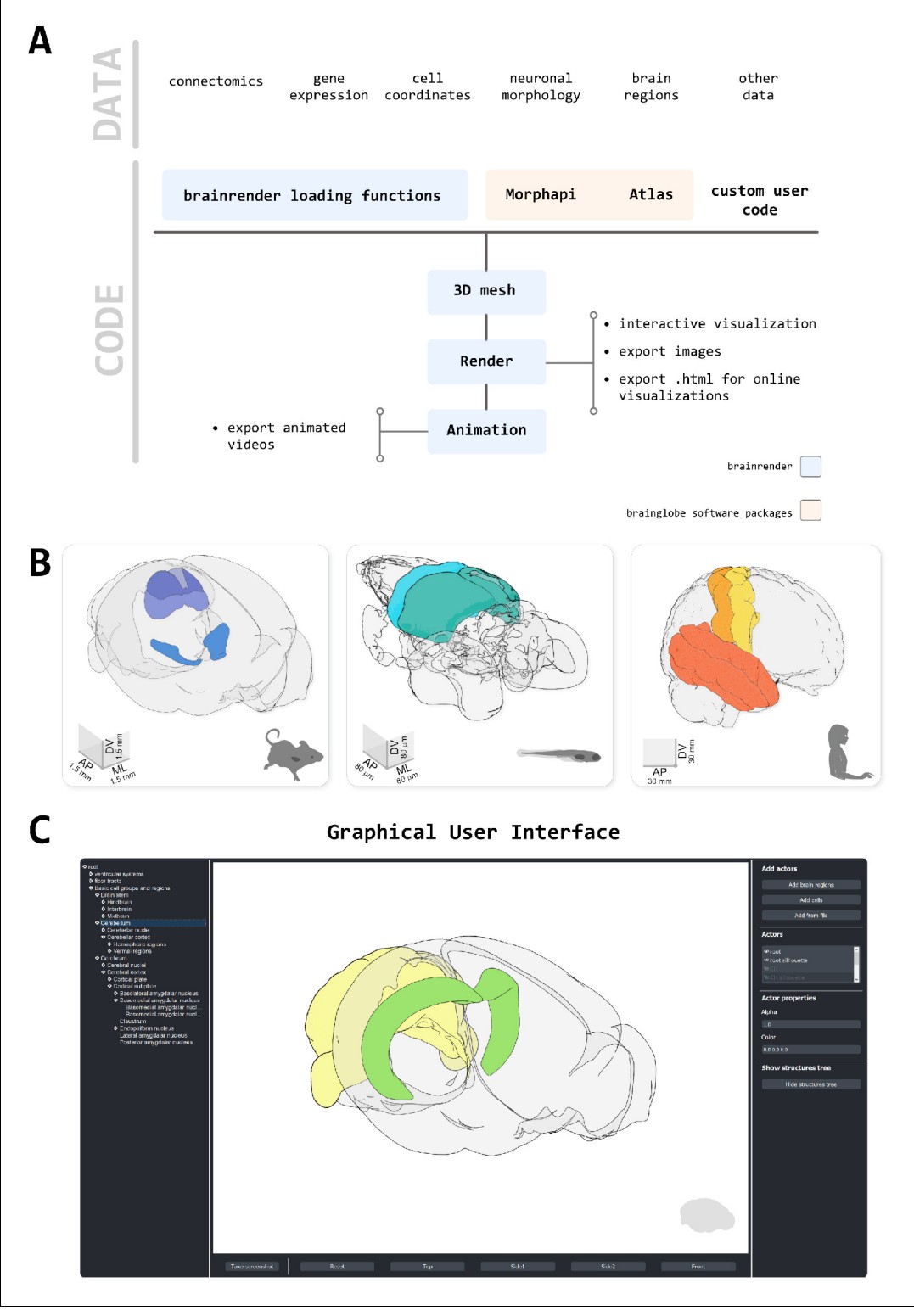

**Figure 1.** Design principles. (**A**) Schematic illustration of how different types of data can be loaded into brainrender using either brainrender's own functions, software packages from the BrainGlobe suite, or custom Python scripts. All data loaded into brainrender is converted to a unified format, which simplifies the process of visualizing data from different sources. (**B**) Using brainrender with different atlases. Visualization of brain atlas data from three different atlases using brainrender. Left, Allen atlas of the mouse brain showing the superficial (SCs) and motor (SCm) subdivisions of the superior colliculus and the Zona Incerta (data from *Wang et al., 2020*).
*Figure 1 continued on next page*

*Figure 1 continued*

Middle, visualization of the cerebellum and tectum in the larval zebrafish brain (data from **Kunst et al., 2019**). Right, visualization of the precentral gyrus, postcentral gyrus, and temporal lobe of the human brain (data from **Ding et al., 2016**). (C) The brainrender GUI. Mouse, human, and zebrafish larvae drawings from scidraw.io (doi. org/10.5281/zenodo.3925991, doi.org/10.5281/zenodo.3926189, doi.org/10.5281/zenodo.3926123).

after processing with dedicated software (e.g., *Tyson et al., 2020a*; *Song et al., 2020*; *Jin et al., 2019*; *Figure 3A*) or as 3D volumetric data (*Figure 3E*). For the latter, brainrender takes care of the conversion of voxels into a 3D mesh for rendering. Furthermore, custom 3D meshes can be created to visualize different types of data. For example, brainrender can import JSON files with tractography connectivity data and create 'streamlines' to visualize efferent projections from a brain region of interested (*Figure 3B*).

Brainrender also simplifies visualizing the location of devices implanted in the brain for neural activity recordings or manipulations, such as electrodes or optical fibers. Post hoc histological images taken to confirm the correct placement of the device can be registered to a reference atlas using appropriate software, and the registered data can be imported into brainrender (*Figure 3C*). This type of visualization greatly facilitates cross-animal comparisons and helps data interpretation within and across research groups.

Finally, brainrender can be used to visualize any object represented by the most commonly used file formats for 3D design (e.g. .obj, .stl), thus ensuring that brainrender can flexibly adapt to the visualization needs of the user (*Figure 3D*).

## Individual neurons and mesoscale connectomics

Recent advances in large field of view and whole-brain imaging allow the generation of brain-wide data at single neuron resolution. Having a platform for visualizing these datasets with ease is critical for exploratory data analyses. Several open-source software packages are available for registering large amounts of such imaging data and automatically identify labeled cells (e.g. expressing fluorescent proteins) (*Tyson et al., 2020a*; *Fürth et al., 2018*; *Goubran et al., 2019*; *Renier et al., 2016*). This processing step outputs a table of coordinates for a set of labeled cells, which can be directly imported into brainrender to visualize a wealth of anatomical data at cellular resolution (*Figure 4A*).

Beyond the location of cell bodies, visualizing the entire dendritic and axonal arbors of single neurons registered to a reference atlas is important for understanding the distribution of neuronal signals across the brain. Single-cell morphologies are often complex 3D structures and therefore poorly represented in 2D images. Generating 3D interactive renderings is thus important to facilitate the exploration of this type of data. Brainrender can be used to parse and render .swc files containing morphological data, and it is fully integrated with morphapi, a software for downloading

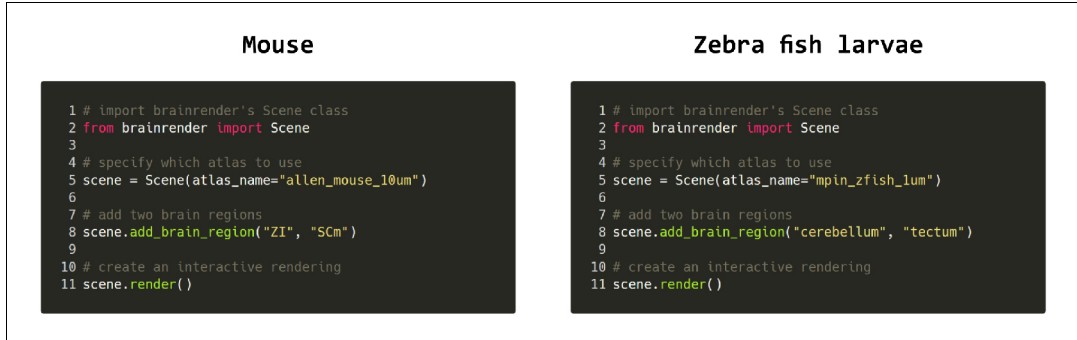

**Figure 2.** Code examples. Example python code for visualizing brain regions in the mouse and larval zebrafish brains. The same commands can be used for both atlases and switching between atlases can be done by simply specifying which atlas to use when creating the visualization. Further examples can be found in brainrender's GitHub repository.

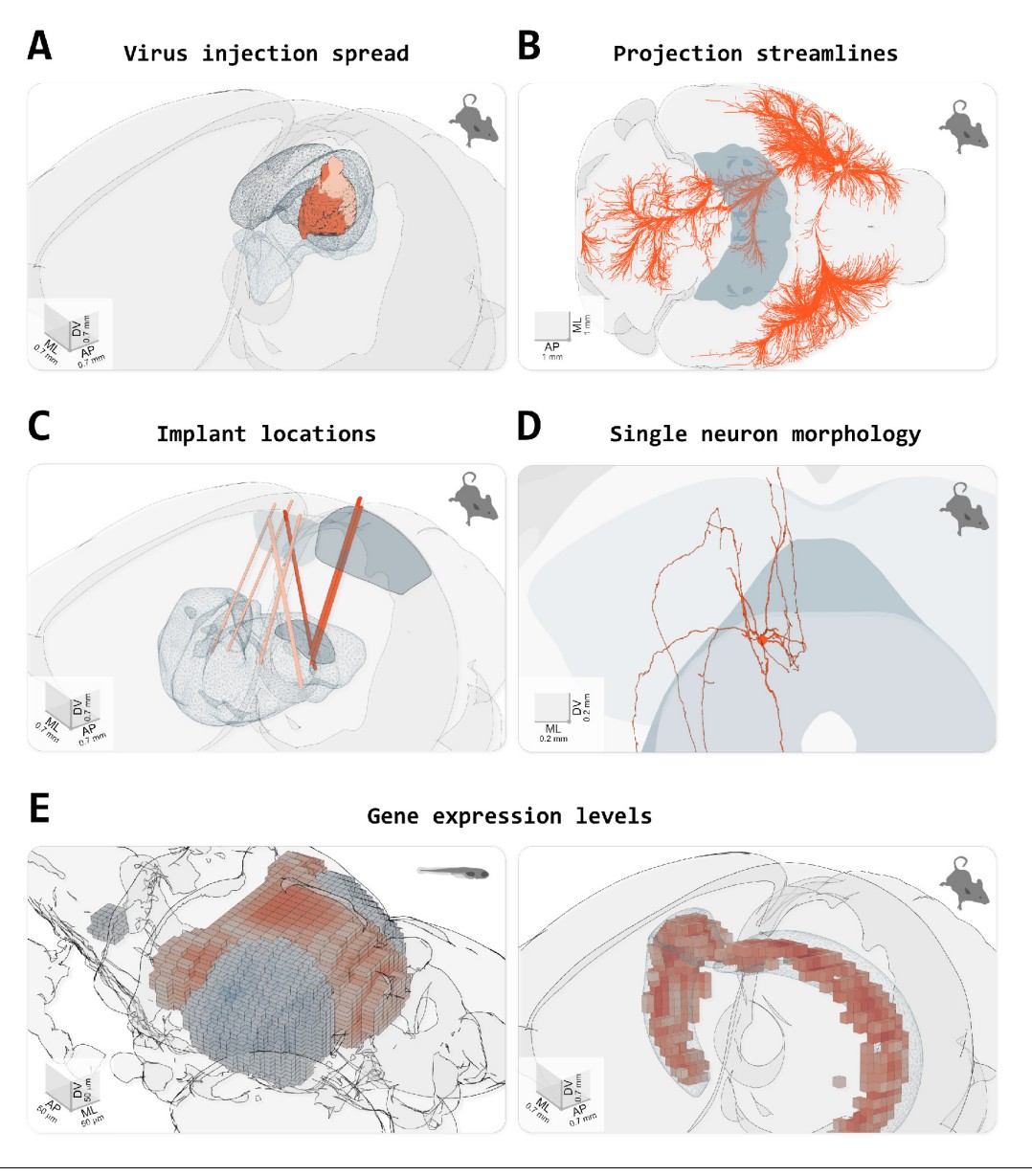

**Figure 3.** Visualizing different types of data in brainrender. (A) Spread of fluorescence labeling following viral injection of AAV2-CRE-eGPF in the superior colliculus of two FLEX-TdTomato mice. 3D objects showing the injection sites were created using custom python scripts following acquisition of a 3D image of the entire brain with serial two-photon tomography and registration of the image data to the atlas' template (with brainreg, *Tyson et al., 2020a*). (B) Streamlines visualization of efferent projections from the mouse primary motor cortex following injection of an anterogradely transported virus expressing fluorescent proteins (original data from *Oh et al., 2014*), downloaded from (Neuroinformatics NL with brainrender). (C) Visualization of the location of several implanted neuropixel probes from multiple mice (data from *Steinmetz et al., 2019*). Dark salmon colored tracks show probes going through both primary/anterior visual cortex (VISp/VISa) and the dorsal lateral geniculate nucleus of the thalamus. (D) Single periaqueductal gray (PAG) neuron. The PAG and superior colliculus are also shown. The neuron's morphology was reconstructed by targeting the expression of fluorescent proteins in excitatory neurons in the PAG via an intersectional viral strategy, followed by imaging of cleared tissue and manual reconstruction of the neuron's morphology with Vaa3D software. Data were registered to the Allen atlas with SHARPTRACK (*Shamash et al., 2018*). The 3D data was saved as a .stl file and loaded directly into brainrender. (E) Gene expression data. Left, expression of genes 'brn3c' and 'nk1688CGt' in the tectum of the larval zebrafish brain (gene expression data from fishatlas.neuro.mpg.de, 3D objects created with custom python scripts). Right, expression of gene 'Gpr161' in the mouse hippocampus (gene expression data from *Wang et al., 2020*),

*Figure 3 continued on next page*

downloaded with brainrender (3D objects created with brainrender). Colored voxels show voxels with high gene expressions. The CA1 field of the hippocampus is also shown.

morphological data from publicly available datasets (e.g. from neuromorpho.org, *Ascoli et al., 2007*; *Figure 4B*).

## Producing figures, videos, and interactive visualizations with brainrender

A core goal of brainrender is to facilitate the production of high-quality images, videos, and interactive visualizations of anatomical data. Brainrender leverages the functionality provided by vedo (*Musy et al., 2019*) to create images directly from the rendered scene. Renderings can also be exported to HTML files to create interactive visualizations that can be hosted online. Finally, functionality is provided to easily export videos from rendered scenes. Animated videos can be created by specifying parameters (e.g. the position of the camera or the transparency of a mesh) at selected keyframes. Brainrender then creates a video by animating the rendering between the keyframes. This approach facilitates the creation of videos while retaining the flexibility necessary to produce

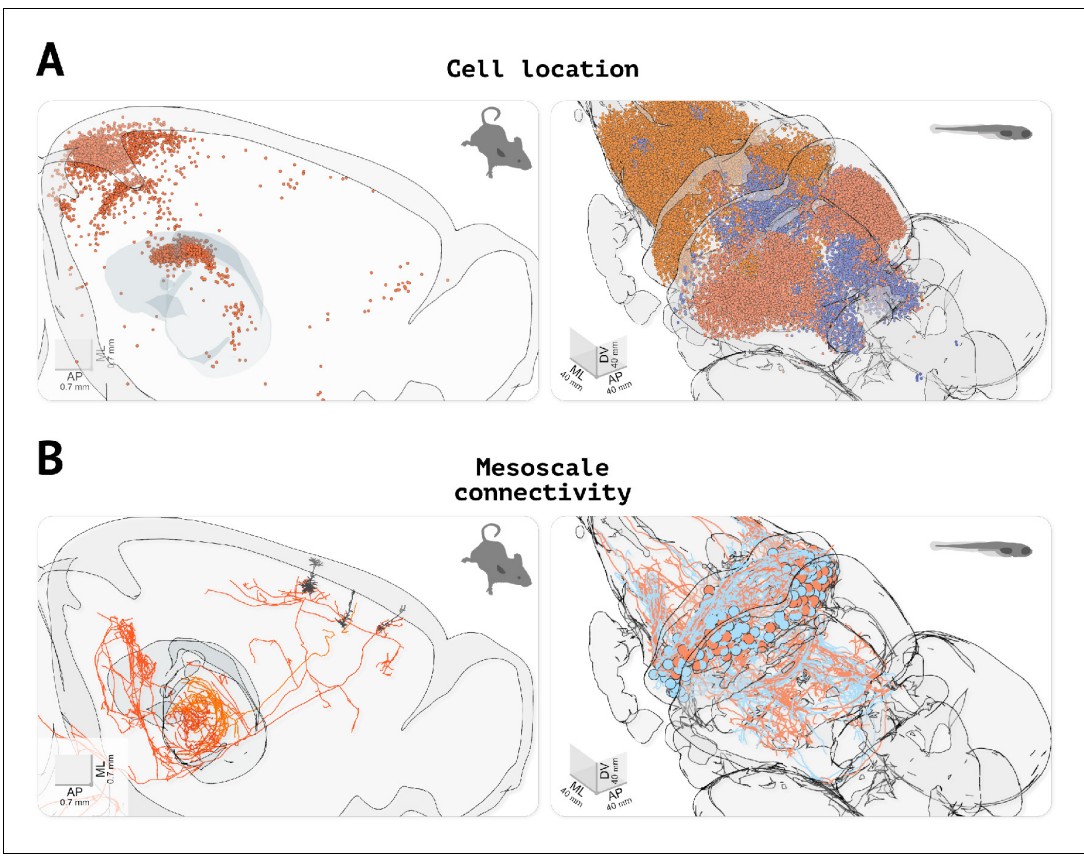

**Figure 4.** Visualizing cell location and morphological data. (**A**) Visualizing the location of labeled cells. Left, visualization of fluorescently labeled cells identified using cellfinder (data from *Tyson and Rousseau, 2020b*). Right, visualization of functionally defined clusters of regions of interest in the brain of a zebrafish larvae during a visuomotor task (data from *Markov et al., 2020*). (**B**) Visualizing neuronal morphology data. Left, three secondary motor cortex neurons projecting to the thalamus (data from *Winnubst et al., 2019*, downloaded with morphapi from neuromorpho.org, *Ascoli et al., 2007*). Right, morphology of cerebellar neurons in larval zebrafish (data from *Kunst et al., 2019*), (downloaded with morphapi). In the left panel of (**A** and **B**), the brain outline was sliced along the midline to expose the data.

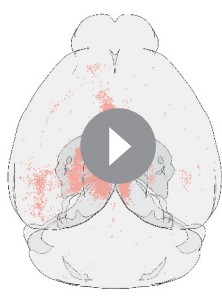

**Video 1.** Example brainrender GUI usage. Short demonstration of how brainrender's GUI can be used to interactively visualize brain regions, labeled cells, and custom meshes.

https://elifesciences.org/articles/65751#video1

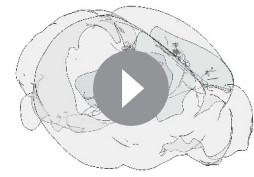

**Video 2.** Animated video created with brainrender. Visualization of neuronal morphologies for two layer 5b pyramidal neurons in the secondary motor area of the mouse brain. *Winnubst et al., 2019*, downloaded with morphapi from neuromorpho.org. The secondary motor area and thalamus are also shown.

https://elifesciences.org/articles/65751#video2

richly animated sequences (*Videos 2–5*). All example figures and videos in this article were generated directly in brainrender, with no further editing.

## Discussion

In this article, we have presented brainrender, a python software for creating 3D renderings of anatomically registered data.

Brainrender addresses the current lack of python-based and user-friendly tools for rendering anatomical data. Being part with BrainGlobe's suite of software tools for the analysis of anatomical data brainrender facilitates the development of integrated analysis pipelines and the re-usability of software tools across model species, minimizing the need for additional software development. Finally, brainrender promises to improve how anatomically registered data are disseminated both in scientific publications and in other media (e.g., hosted online).

### Limitations and future directions

With brainrender, we aimed to make the rendering process as simple as possible. Nevertheless, some more technically demanding pre-processing steps of raw image data are necessary before they can be visualized in brainrender. In particular, a critical step for visualizing anatomical data is the registration to a reference template (e.g., one of the atlases provided by the AtlasAPI). While this step can be challenging and time-consuming, the brainglobe suite provides software to facilitate this process (e.g., brainreg and bg-space), and alternative software tools have been developed before for this purpose (e.g., *Song et al., 2020*; *Jin et al., 2019*). Additional information about data registration can be found in brainglobe's and brainrender's online documentation, as well as in the examples in brainrender's GitHub repository. A related challenge is integrating new anatomical atlases into the AtlasAPI. While we anticipate that most users will not have this need, it is a non-trivial task that requires considerable programming skills. We believe that brainglobe's AtlasAPI greatly facilitates this process, which is presented in *Claudi et al., 2020* and has extensive online documentation.

Brainrender has been optimized for rendering quality instead of rendering performance. Other commonly used software tools like napari (*Sofroniew and Lambert, 2020*) and ImageJ are dedicated to visualizing N-dimensional image data and perform very well even on large datasets. When comparing brainrender with other software, it is important to note brainrender is intended to work primarily with mesh data and not 3D image data. Although it can display image data (e.g., with the Volume actor), this functionality is not as fully developed as that using mesh data. A direct benchmarking comparison between brainrender and napari shows that brainrender is 5× slower than napari at visualizing image data, but 20× faster at visualizing mesh data. In both cases, however, brainrender achieves superior rendering quality. Other software packages dedicated to high-

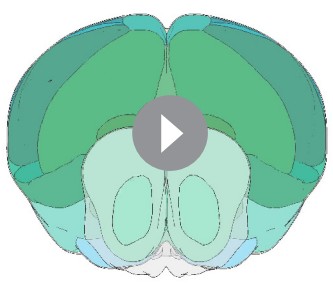

**Video 3.** Animated video created with brainrender. Frontal view of all brain regions in the Allen Mouse Brain atlas as the brain is progressively 'sliced' in the rostro-caudal direction.

https://elifesciences.org/articles/65751#video3

performance rendering, such as Blender, can handle mesh data with a performance that surpasses brainrender. Their use, however, comes with the large overhead of learning a very complex software to generate what most often will be simple renderings. It also requires that the users themselves take care of downloading, storing, and accessing mesh data from the anatomical atlases. Nevertheless, the rendering performance of brainrender could be a target for improvement in future versions, both for images and for mesh data, through optimizing the Actor classes. While we have designed brainrender usage to require minimal programming expertise, installing python and brainrender may still prove challenging for some users. In the future, we aim to make brainrender a stand-alone application that can be simply downloaded and locally installed, either through Docker containers or through executable files. Further possible improvements include the development of plug-ins for loading of data from file formats other than those already supported, and improvements to the GUI functionality. Moreover, in addition to images and videos, brainrender can be used to export renderings as HTML files and generate online 3D interactive renderings. Currently, however, embedding renderings into a web page remains far from a trivial task. Further developments on this front should make it possible to easily host interactive renderings online, therefore improving how anatomically registered data are disseminated both in scientific publications and in other media. While we plan to continue developing brainrender in the future, we welcome contributions from the community. Users should feel encouraged to contribute irrespective of their programming experience, and we note that the programming ability of many biologists is often better than what they perceive it to be. We especially welcome contributions aimed at improving the user-experience of brainrender, at any level of interaction. Contributions can involve active development of brainrender's code base, but they can also be bug reports, features request, improvements with the online documentation, and help answering users' questions.

# Materials and methods

## Key resources table

| Reagent type (species) or resource | Designation | Source or reference | Identifiers | Additional information |
|---|---|---|---|---|
| Software, algorithm | Numpy | https://doi.org/10.1038/s41586-020-2649-2 | RRID:SCR_008633 | |
| Software, algorithm | Vtk | https://doi.org/10.1016/j.softx.2015.04.001 | RRID:SCR_015013 | |
| Software, algorithm | Vedo | https://zenodo.org/record/4287635 | | |
| Software, algorithm | BrainGlobe Atlas API | https://doi.org/10.21105/joss.02668 | | |
| Software, algorithm | Pandas | https://doi.org/10.5281/zenodo.3509134 | | |
| Software, algorithm | Matplotlib | doi: 10.1109/MCSE.2007.55 | RRID:SCR_008624 | |
| Software, algorithm | Jupyter | doi:10.3233/978-1-61499-649-1-87 | RRID:SCR_018416 | |

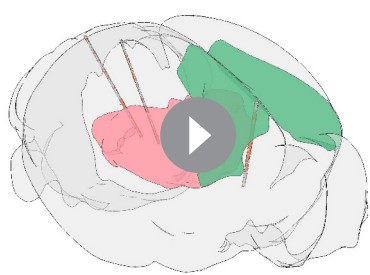

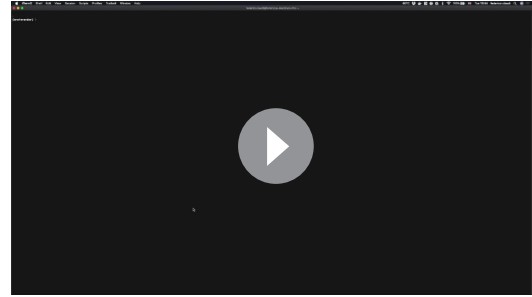

**Video 4.** Animated video created with brainrender. Visualization of the location of three implanted neuropixel probes from multiple mice (data from *Steinmetz et al., 2019*). Every 0.5 s, a subset of the probes' electrodes that detected a neuron's action potential are shown in salmon to visualize neuronal activity.

https://elifesciences.org/articles/65751#video4

**Video 5.** Animated video created with brainrender showing the location of cells labeled by targeted expression of a fluorescent protein identified with cellfinder (data from *Tyson et al., 2020a*). In dark blue: streamline visualization of efferent projections from the retrosplenial cortex following injection of an anterogradely transported virus expressing fluorescent proteins (data from *Oh et al., 2014*).

https://elifesciences.org/articles/65751#video5

## Brainrender's workflow

Brainrender is written in Python three and depends on standard python packages such as numpy, matplotlib, and pandas (*Harris et al., 2020*; *Hunter, 2007*; *The pandas development team, 2020*) and on vedo (*Musy et al., 2019*) and Brain-Globe's AtlasAPI (*Claudi et al., 2020*). Extensive documentation on how to install and use brainrender can be found at docs.brainrender.info, and we provide here a only brief overview of the workflow in brainrender. The GitHub repository also contains detailed examples of Python scripts and Jupyter notebooks (*Kluyver et al., 2016*). All brainrender's code is open-source and has been deposited in full in the GitHub repository and at PyPI (a repository of Python software) under a permissive BSD 3-Clause license. We welcome any user to download and inspect the source code, modify it as needed, or contribute to brainrender's development directly.

Brainrender can be installed in any python environment using python version $\geq$ 3.6.0. We recommend the creation of an anaconda or virtual environment with an appropriate python version for use with brainrender. Installing brainrender is then as simple as 'pip install brainrender' although additional optional packages might have to be installed separately (e.g., to access data from the Allen Institute).

The central element of any visualization produced by brainrender is the Scene. A Scene controls which elements (Actors) are visualized and coordinates the rendering, the position of the camera's point of view, the generation of screenshots and animations from the rendered scene, and other important actions.

Actors can be added to the scene in several ways. When loading data directly from a file with 3D mesh information (e.g. .obj), an Actor is generated automatically to represent the mesh in the rendering. When rendering data from other sources (e.g. from a .swc file with neuronal morphology or from a table of coordinates of labeled cells), dedicated functions in brainrender parse the input data and generate the corresponding Actors. Actors in brainrender have properties, such as color and transparency, that can be used to specify the appearance of a rendered actor accordingly to the

**Table 1.** Machine configurations used for benchmark tests.

| N | OS | CPU | GPU |
|---|---|---|---|
| 1 | Macos Mojave 10.14.6 | 2.3 ghz Intel Core i9 | Radeon Pro 560 × 4 GB GPU |
| 2 | Ubuntu 18.04.2 LTS x86 64 | Intel i7-8565U (x) @ 4.5 ghz | NO GPU |
| 3 | Windows 10 | Intel(R) Core i7-7700HQ 2.8 ghz | NO GPU |
| 4 | Windows 10 | Intel(R) Xeon(R) CPU E5-2643 v3 3.4 ghz | NVIDIA geforce GTX 1080 Ti |

**Table 2.** Benchmark tests results.

The number of actors refers to the total number of elements rendered, and the number of vertices refers to the total number of mesh vertices in the rendering.

| Test | Machine | GPU | # actors | # vertices | FPS | Run duration |
|---|---|---|---|---|---|---|
| 10 k cells | 1 | Yes | 3 | 1,029,324 | 24.76 | 0.81 |
| | 2 | No | 3 | 1,029,324 | 22.46 | 1.16 |
| | 3 | No | 3 | 1,029,324 | 20.00 | 1.41 |
| | 4 | Yes | 3 | 1,029,324 | 100.00 | 1.34 |
| 100 k cells | 1 | Yes | 3 | 9,849,324 | 18.87 | 3.23 |
| | 2 | No | 3 | 9,849,324 | 14.91 | 4.34 |
| | 3 | No | 3 | 9,849,324 | 0.43 | 7.94 |
| | 4 | Yes | 3 | 9,849,324 | 1.20 | 1.13 |
| 1 M cells | 1 | Yes | 3 | 98,049,324 | 2.65 | 31.01 |
| | 2 | No | 3 | 98,049,324 | 2.55 | 96.49 |
| | 3 | No | 3 | 98,049,324 | 0.03 | 86.75 |
| | 4 | Yes | 3 | 9,8049,324 | 0.13 | 36.57 |
| Slicing 10 k cells | 1 | Yes | 3 | 237,751 | 37.64 | 0.96 |
| | 2 | No | 3 | 237,751 | 39.10 | 1.25 |
| | 3 | No | 3 | 237,751 | 26.32 | 1.88 |
| | 4 | Yes | 3 | 237,751 | 200.00 | 1.34 |
| Slicing 100 k cells | 1 | Yes | 3 | 276,092 | 31.79 | 7.77 |
| | 2 | No | 3 | 276,092 | 25.98 | 9.09 |
| | 3 | No | 3 | 276,092 | 21.28 | 16.88 |
| | 4 | Yes | 3 | 276,092 | 111.11 | 9.65 |
| Slicing 1 M cells | 1 | Yes | 3 | 275,069 | 11.23 | 91.31 |
| | 2 | No | 3 | 275,069 | 5.39 | 104.79 |
| | 3 | No | 3 | 275,069 | 5.03 | 158.99 |
| | 4 | Yes | 3 | 275,069 | 37.04 | 97.43 |
| Brain regions | 1 | Yes | 1678 | 1,864,388 | 9.38 | 11.78 |
| | 2 | No | 1678 | 1,864,388 | 7.61 | 27.40 |
| | 3 | No | 1678 | 1,864,388 | 6.49 | 46.79 |
| | 4 | Yes | 1678 | 1,864,388 | 11.90 | 35.83 |
| Animation | 1 | Yes | 8 | 96,615 | 9.91 | 18.98 |
| | 2 | No | 8 | 96,615 | 22.12 | 12.63 |
| | 3 | No | 8 | 96,615 | 15.15 | 11.92 |
| | 4 | Yes | 8 | 96,615 | 47.62 | 12.29 |
| Volume | 1 | Yes | 12 | 49,324 | 1.79 | 2.31 |
| | 2 | No | 12 | 49,324 | 1.66 | 1.95 |
| | 3 | No | 12 | 49,324 | 3.55 | 2.15 |
| | 4 | Yes | 12 | 49,324 | 23.26 | 1.21 |

user's aesthetic preferences. Brainrender's Scene and Actor functionality use vedo as the rendering engine (GitHub repository; *Musy et al., 2019*).

In addition to data loaded from external files, brainrender can directly load atlas data containing, for example, the 3D meshes of individual brain regions. This is done via BrainGlobe's AtlasAPI to allow the same programming interface in brainrender to visualize data from any atlas supported by the AtlasAPI. Brainrender also provides additional functionality to interface with data available from

projects that are part of the Allen Institute Mouse Atlas and Mouse Connectome projects (*Wang et al., 2020*; *Oh et al., 2014*). These projects provide an SDK (Software Development Kit) to directly download data from their database, and brainrender provides a simple interface for downloading gene expression and connectomics (streamlines) data. All atlas and connectomics data downloaded by brainrender can be loaded directly into a Scene as Actors.

Visualizing morphological data with reconstructions of individual neurons can be done by loading these type of data directly from .swc files or by downloading them in Python using morphapi – software from the BrainGlobe suite that provides a simple and unified interface with several databases of neuron morphologies (e.g., neuromorpho.org, *Ascoli et al., 2007*). Data downloaded with morphapi can be loaded directly into a brainrender scene for visualization.

## Example code

As a demonstration of how easily renderings can be created in brainrender, the Python code (*Figure 5*) illustrates how to create a Scene and add Actors by loading 3D data from an .obj file and then adding brain regions to the visualization. Brainrender's GitHub repository provides several simple and concise examples about how to use brainrender to load user data, atlas data, to edit rendered meshes (e.g., to change color or cut them with a plane), to save screenshots from rendered scenes, and to create animated videos.

While brainrender is intended to be mainly a visualization tool, simple analyses can be carried out directly by leveraging functionality from either vedo or BrainGlobe's AtlasAPI. For example, Vedo can access properties of actors added to a brainrender scene, which could be used to measure the distance between two actors or to check if two actors' meshes intersect (*Figure 6A*). Similarly, BrainGlobe's AtlasAPI provides methods to, for example, check whether a point (defined by a set of coordinates) is contained in a brain region of interest or to retrieve brain regions that are above or below a brain region of interest in the atlas' hierarchy (*Figure 6B*).

The code and data used to generate the figures and videos in this article are made freely available at brainrender's GitHub repository and provides examples of more advanced usage of brainrender's functionality.

**Figure 5.** Code examples. (**A**) Example code to visualize a set of labeled cells coordinates using the Points actor class. (**B**) Code example illustrating how to override brainrender's default settings and how to use custom camera settings. (**C**) Code example showing how custom mesh objects saved as .obj and .stl files can be visualized in brainrender. (**D**) Example usage of brainrender's Animation class to create custom animations. Further examples can be found in brainrender's GitHub repository.

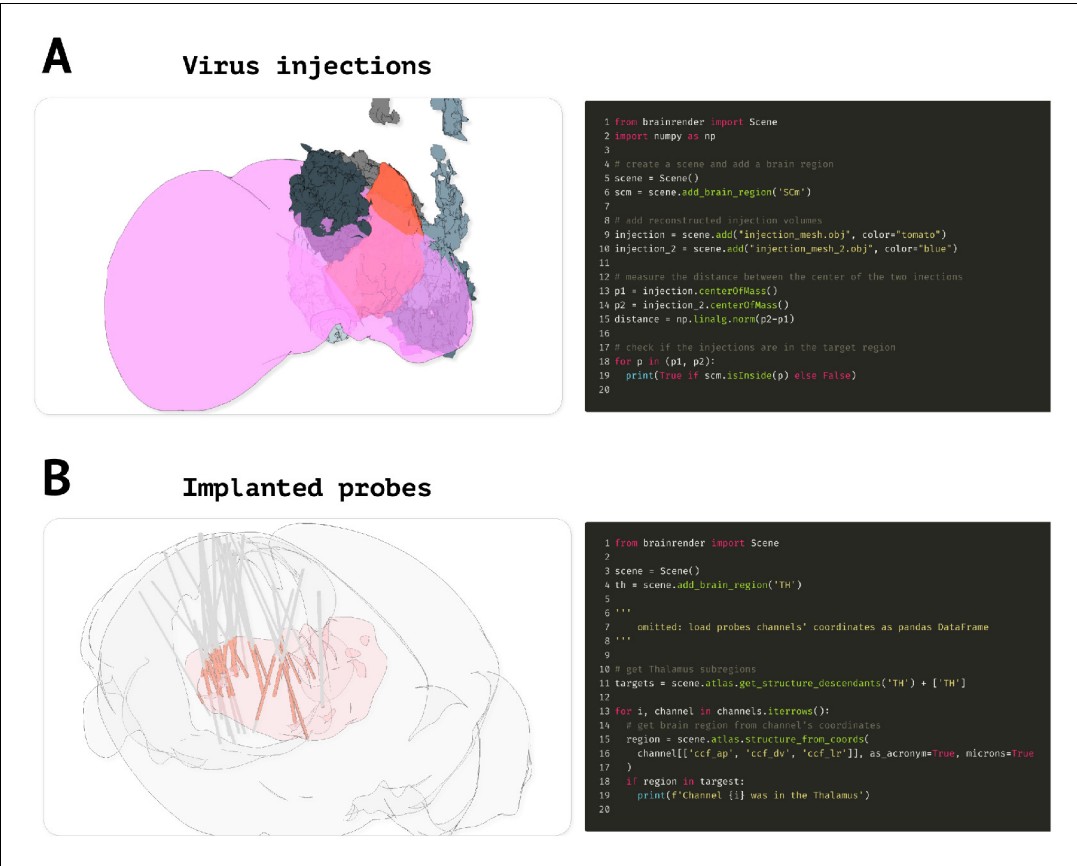

**Figure 6.** Advanced code examples. (**A**) Example code to measure the distance between actors and if a given actor is contained in a target brain region. Left: virus injection volumes (red and gray) reconstructed from virus injections targeted at the superior colliculus (magenta). Gray colored injection volumes show data from the Allen Mouse Connectome *Oh et al., 2014*. Right: example code to measure the distance between the center of two brainrender actors and to check if an actor's center is contained in a brain region of interest. (**B**) Code example illustrating how check if a point (e.g., representing a labeled cell) is in a brain region of interest. Left: visualization of reconstructed probe positions from several individual animals, data from *Steinmetz et al., 2019*. Probe channels located in the thalamus (red) are highlighted. Right: example code showing how to use BrainGlobe's AtlasAPI to verify whether a point (here representing a probe channel) is contained in a brain region of interest or any of its substructures. Further examples can be found in brainrender's GitHub repository.

## Benchmark tests

We designed a series of benchmark tests aimed at evaluating brainrender's performance with different combinations of hardware and operating system. We used five tests designed to cover most aspects of brainrender's functionality:

- rendering large numbers ($1^4$, $1^6$, $1^7$) of cells using the Points actor.
- using a plane to 'slice' the same number of cells (using the Scene.slice method).
- rendering more than 1000 individual meshes representing brain regions from the Allen institute's mouse brain.
- making a short (3 s, 10 fps) animation of a spinning brain with several brain regions' meshes displayed.
- rendering (10 times) a 3D image representing the voxel-wise expression levels of gene Gpr161 in the mouse brain (data from the Allen Institute).

For each test, we estimated the time necessary to complete the test script as well as the frame rate of the interactive rendering. Four machines were used for benchmark tests (see *Table 1*). The results of the benchmark tests (see Key resource table) illustrate that although a GPU improves performance, in the absence of a dedicated GPU brainrender can handle rich interactive visualizations

(for most user cases, the number of rendered mesh vertices is much lower than that used in the tests).

## Acknowledgements

We thank Yu Lin Tan for sharing the single neuron morphology shown in 3D. The illustrations of a human, mouse, and zebrafish used in *Figures 1*, *2,* and *3* were obtained from scidraw.io.

## Additional information

### Funding

| Funder | Grant reference number | Author |
|---|---|---|
| Gatsby Charitable Foundation | GAT3361 | Troy W Margrie Tiago Branco |
| Wellcome | 214333/Z/18/Z | Troy W Margrie |
| Wellcome | 214352/Z/18/Z | Tiago Branco |
| Wellcome | 090843/F/09/Z | Troy W Margrie Tiago Branco |
| Deutsche Forschungsge-meinschaft | 390857198 | Ruben Portugues |

The funders had no role in study design, data collection and interpretation, or the decision to submit the work for publication.

### Author contributions

Federico Claudi, Conceptualization, Resources, Software, Validation, Visualization, Methodology, Writing - original draft, Project administration, Writing - review and editing; Adam L Tyson, Luigi Petrucco, Conceptualization, Resources, Software, Writing - original draft; Troy W Margrie, Ruben Portugues, Supervision, Project administration; Tiago Branco, Supervision, Funding acquisition, Writing - original draft, Project administration, Writing - review and editing

### Author ORCIDs

Adam L Tyson (iD) https://orcid.org/0000-0003-3225-1130
Troy W Margrie (iD) http://orcid.org/0000-0002-5526-4578
Ruben Portugues (iD) http://orcid.org/0000-0002-1495-9314
Tiago Branco (iD) https://orcid.org/0000-0001-5087-3465

### Decision letter and Author response
Decision letter https://doi.org/10.7554/eLife.65751.sa1
Author response https://doi.org/10.7554/eLife.65751.sa2

## Additional files

### Supplementary files
• Transparent reporting form

### Data availability

All code has been deposited on GitHub and is freely accessible (https://github.com/brainglobe/brainrender).

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
