## [Decision Letter]

**Acceptance summary:**

Claudi et al., present a new tool for visualizing brain maps. In the era of new technologies to clear and analyze brains of model organisms, new tools are becoming increasingly important for researchers to interact with this data. Here, the authors report on a new python-based tool for just this: exploring, visualizing, and rendering this high dimensional (and large) data. Moreover, the authors provide rendering performance benchmarking, open source code, and extensive documentation. We believe this tool will be of great interest to researchers who need to visualize multiple brains within several key model organisms, and is written such that it can be adopted rapidly by the neuroscientific community.

**Decision letter after peer review:**

Thank you for submitting your article "Brainrender: a python-based software for visualizing anatomically registered data" for consideration by *eLife*. Your article has been reviewed by three peer reviewers, one of whom is a member of our Board of Reviewing Editors, and the evaluation has been overseen by Kate Wassum as the Senior Editor. The following individual involved in review of your submission has agreed to reveal their identity: Juan Nunez-Iglesias (Reviewer #2).

Essential Revisions:

1) All reviews highlight some additional information required regarding usability (which file types are already supported out of the box, computational resources, run times, minimal example code in methods, how hard/easy is it given one's level of coding, etc.). We agreed these are critical. Please address each of these concerns.

2) 2 of 3 reviewers mention the need for citations of dependencies, so please address both reviewer #1 and #2's specific comments on this below.

3) Lastly, please consider revising the Conclusion/Discussion section. Some text feels redundant, and this space could be used to discuss limitations and future expansions more directly. Each review highlights some very nice strengths and some limitations, so please consider the recommendations when you edit this section.

Please include a key resource table if you have not already done so.

*Reviewer #2 (Recommendations for the authors):*

The paper does a great job of describing the need for this new software, as well most of the capabilities of the software. I did wonder specifically which file formats, other than.obj and.stl, were supported. Specifically, in subsection “Design principles and implementation”, it would be good to introduce at least one sentence about the exact IO capabilities of *brainrender* “out of the box”, as well as noting that the full scientific Python ecosystem can provide support for other formats. In fact, a whole section/paragraph about how to import data from custom file formats (e.g. a.czi file?) would be most welcome. I expect the first question of many readers will be "how do I get my data into this to try it out?"

The whole Conclusions section felt rather redundant. Even though this is common in scientific writing, I'd remove most of it, reduce it to one sentence, and instead elaborate on "limitations and future directions". I expect the authors can certainly come up with more ideas there (e.g. IO plugins?), as well as existing bugs (e.g. Video 4 has some distracting flickering on the electrodes – it's unclear to me whether this is expected, but I suspect not). Additionally, I recommend using this section to point out that readers can and should get involved with future development. The line "We welcome any user to.…" should be a paragraph in future directions rather than a one-liner in the Materials and methods. "We welcome users to submit bug reports and feature requests on our GitHub issues page, as well as usage questions on image.sc. Further, given the completely open source nature of the software, we especially welcome users who would like to help us improve the software for their use case…" In my experience, most biologists don't see themselves as capable of doing this, and most of those are wrong, so this is a good space to dispel that notion.

As an aside, the references need work. Many are incomplete, including two from this paper's co-authors (Tyson 2020a/b). Additionally, although NumPy is appropriately cited, several other packages are not:

– matplotlib is used for its colormaps; this should be mentioned in the methods and matplotlib cited.

– Jupyter is used for documentation and the Scene has some Jupyter compatibility, but Jupyter is not cited. Citation info for Jupyter appears to be discussed in this github comment: https://github.com/jupyter/jupyter/issues/190#issuecomment-721264013

– Pandas is used but not cited.

– I notice napari is used but only for its theme – I'd say it's appropriate to not cite it. Indeed, I would recommend the authors remove that dependency and instead copy the very small amount of information that they need from it.

A good resource for citation info in the scientific Python ecosystem is found here:

https://www.scipy.org/citing.html

I would also recommend naming NumPy specifically in the Materials and methods section (as well as the above packages), as I initially missed that the authors had appropriately cited it – just buried under the generic "standard python packages"

Finally, although there is a screenshot of the *brainrender* UI, *and* the supplementary videos show renderings created with *brainrender*, there is not a screen-captured video demonstrating the *brainrender* UI being used to generate a video. I would suggest including one as part of the supplementary materials, as that is something that many readers will be looking for – it is very hard to convey the usability of GUI software using text alone.

All in all, those are all easily fixed points and would encourage publication of the paper.

*Reviewer #3 (Recommendations for the authors):*

1) Emphasize the importance and difficulty of having accurately registered data. In our experience, this is the hardest part of the process, and it is just lightly discussed as a requirement for using the tool.

2) They state that it is easy to incorporate another atlas through the brainreg software, which can then be used with *brainrender*. As mentioned, it is our opinion that this is not a straight-forward task and that it would require significant programming skills to implement. Please provide more direction about how this can be done and what the constraints are.

3) One of the stated advantages of this software is the ability to visualize multiple types of data and data from sources that are external to the atlas generators. Such visualizations can potentially be used to reveal consistency and/or novelty across data types. However, ultimately you would want to be able to measure these differences. Including examples about how to extract and compare features across data types and then visualize those differences with *brainrender* would strengthen the paper.

4) The snapshots of code presented as figures don't add much to the manuscript. Consider highlighting how-to videos instead.

5) As stated in earlier comments, some of the functionality is not clear, particularly for people unfamiliar or new to 3D visualization or coding in general. Even experienced developers would benefit from specs for the various input data types, for example. A little more explanation, particularly in cases where the interface is different (such as with the helper functions used to make some actors), can greatly improve the user experience.

6) Though the purpose of this tool is not to develop a registration algorithm for anatomical reference atlases, or to perform data analysis, we view these steps as the most difficult and necessary steps in this process. 3D rendered data visualizations are informative, but not meaningful without accurate registration to begin with and without quantitative analysis to back it up. As they point out in the introduction, other, similar tools (natverse, MagellanMapper) have both visualization and analysis capabilities.

---

## [Author Response]

Essential Revisions:1) All reviews highlight some additional information required regarding usability (which file types are already supported out of the box, computational resources, run times, minimal example code in Materials and methods, how hard/easy is it given one's level of coding, etc.). We agreed these are critical. Please address each of these concerns.2) 2 of 3 reviewers mention the need for citations of dependencies, so please address both reviewer #1 and #2's specific comments on this below.3) Lastly, please consider revising the Conclusion/Discussion section. Some text feels redundant, and this space could be used to discuss limitations and future expansions more directly. Each review highlights some very nice strengths and some limitations, so please consider the recommendations when you edit this section.Please include a key resource table if you have not already done so.

We are grateful for the reviewers’ comments, which we found very helpful for improving the manuscript, as well as some of the code base and online documentation. We have addressed all of the concerns raised by including information about usability where this was missing, running benchmarking tests, citing references relevant to brainrender’s dependencies, and extensively revising the Discussion section to focus on the software’s strengths, limitations and opportunities for future improvements.

We now also include a key resource table.

Reviewer #2 (Recommendations for the authors):The paper does a great job of describing the need for this new software, as well most of the capabilities of the software. I did wonder specifically which file formats, other than.obj and.stl, were supported. Specifically, in subsection “Design principles and implementation”, it would be good to introduce at least one sentence about the exact IO capabilities of brainrender *out of the box*, as well as noting that the full scientific Python ecosystem can provide support for other formats. In fact, a whole section/paragraph about how to import data from custom file formats (e.g. a.czi file?) would be most welcome. I expect the first question of many readers will be "how do I get my data into this to try it out?"

Yes, this is a very good point and we have now added a whole new section to the online documentation detailing the available options for getting data into *brainrender*: https://docs.brainrender.info/usage/using-your-data. We have also added new examples to the online repository illustrating how users can load and re-orient their data to visualize them in *brainrender* alongside atlas data.

In brief, *brainrender* can load directly cell coordinates and image (volume) data from.npy files, it can load streamlines data from.json (provided that they are in the correct format, as now stated in the online documentation) and it can load neuron morphology data from.swc (using morphapi). As suggested by the reviewer, we now include this information in the manuscript. We now also point out that the wider python ecosystem provides libraries for loading the most used file formats, and that *brainglobe* provides *Imio* as a convenient tool for loading anatomical data.

In the revised manuscript we also emphasize the critical data pre-processing steps that must be done before using *brainrender* (e.g., registration to an atlas template). As pointed out by another reviewer, these can be challenging for some users, and thus we now also briefly mention *brainreg* and *bg-space* as two *brainglobe’s* suite tools that can facilitate these preprocessing steps.

The whole Conclusions section felt rather redundant. Even though this is common in scientific writing, I'd remove most of it, reduce it to one sentence, and instead elaborate on "limitations and future directions". I expect the authors can certainly come up with more ideas there (e.g. IO plugins?), as well as existing bugs (e.g. Video 4 has some distracting flickering on the electrodes – it's unclear to me whether this is expected, but I suspect not). Additionally, I recommend using this section to point out that readers can and should get involved with future development. The line "We welcome any user to.…" should be a paragraph in future directions rather than a one-liner in the Materials and methods. "We welcome users to submit bug reports and feature requests on our GitHub issues page, as well as usage questions on image.sc. Further, given the completely open source nature of the software, we especially welcome users who would like to help us improve the software for their use case…" In my experience, most biologists don't see themselves as capable of doing this, and most of those are wrong, so this is a good space to dispel that notion.

We have taken the reviewer’s advice and substituted the Conclusions section by an expanded subsection “Limitations and future direction”, which now discusses additional topics such as rendering performance and processing steps required before data can be used meaningfully in *brainrender*.

Incidentally, the “flickering” observed in Video 4 was not a bug: it was meant to show which probe channels detected a spike at any given moment in time (we now realise that the video legends were not appended to the video; we apologise for this and have included video legends in a separate submission file). We have now edited the video so that the highlighted channels are updated every 5seconds (instead of every frame) hopefully improving the video’s quality.

We strongly agree that development should be open to all and in particular to the end users who benefit the most from improvements in the software. We also very much agree with the comment on the perceived programming ability of most biologists. As suggested by the reviewer we have emphasized this point in the discussion and explicitly invited users to contribute regardless of their programming experience. We now write:

“While we plan to continue developing *brainrender* in the future, we welcome contributions from the community. Users should feel encouraged to contribute irrespective of their programming experience, and we note that the programming ability of many biologists is often better than what they perceive it to be. We especially welcome contributions aimed at improving the user-experience of *brainrender*, at any level of interaction. Contributions can involve active development of *brainrender's* code base, but they can also be bug reports, features request, improvements with the online documentation and help answering users' questions.”

As an aside, the references need work. Many are incomplete, including two from this paper's co-authors (Tyson 2020a/b). Additionally, although NumPy is appropriately cited, several other packages are not:– matplotlib is used for its colormaps; this should be mentioned in the methods and matplotlib cited.– Jupyter is used for documentation and the Scene has some Jupyter compatibility, but Jupyter is not cited. Citation info for Jupyter appears to be discussed in this github comment: https://github.com/jupyter/jupyter/issues/190#issuecomment-721264013– Pandas is used but not cited.– I notice napari is used but only for its theme – I'd say it's appropriate to not cite it. Indeed, I would recommend the authors remove that dependency and instead copy the very small amount of information that they need from it.A good resource for citation info in the scientific Python ecosystem is found here:https://www.scipy.org/citing.htmlI would also recommend naming NumPy specifically in the methods section (as well as the above packages), as I initially missed that the authors had appropriately cited it – just buried under the generic "standard python packages"

Thank you for pointing these out, we apologize for these shortcomings which we have now corrected. As suggested by the reviewer we have also updated *brainrender*’s code to remove the dependency on *napari*.

Finally, although there is a screenshot of the brainrender UI, *and* the supplementary videos show renderings created with brainrender, there is not a screen-captured video demonstrating the brainrender UI being used to generate a video. I would suggest including one as part of the supplementary materials, as that is something that many readers will be looking for – it is very hard to convey the usability of GUI software using text alone.

We have added a supplementary video (Video 5) illustrating the main functionality supported by the GUI.

All in all, those are all easily fixed points and would encourage publication of the paper.

Thank you for the support and for raising these points.

Reviewer #3 (Recommendations for the authors):1) Emphasize the importance and difficulty of having accurately registered data. In our experience, this is the hardest part of the process, and it is just lightly discussed as a requirement for using the tool.

We have expanded the discussion on data registration in subsection “Limitations and futured directions” of the revised manuscript:

“With *brainrender* we aimed to make the rendering process as simple as possible. Nevertheless, some more technically demanding pre-processing steps of raw image data are necessary before they can be visualized in *brainrender*. In particular, a critical step for visualizing anatomical data is the registration to a reference template (e.g., one of the atlases provided by the AtlasAPI). While this step can be challenging and time consuming, the brainglobe suite provides software to facilitate this process (e.g., brainreg and bg-space), and alternative software tools have been developed before for this purpose (e.g., Song et al., (2020), Jin et al., (2019)). Additional information about data registration can be found in brainglobe's and brainrender's online documentation, as well as in the examples in brainrender GitHub repository.”

In addition, we have added a new section to the online documentation detailing how users can visualize their data in brainrender: https://docs.brainrender.info/usage/using-your-data, and we have added new examples to the online repository illustrating how users can load and re-orient their data to visualize them in *brainrender* alongside atlas data.

2) They state that it is easy to incorporate another atlas through the brainreg software, which can then be used with brainrender. As mentioned, it is our opinion that this is not a straight-forward task and that it would require significant programming skills to implement. Please provide more direction about how this can be done and what the constraints are.

We take the reviewer’s point and we have now removed the wording “new atlases can be easily adapted to work with the API” from the Results section. We have also edited the Discussion to directly address this issue:

“A related challenge is integrating new anatomical atlases into the AtlasAPI. While we anticipate that most users will not have this need, it is a non-trivial task that requires considerable programming skills. We believe that brainglobe’s AtlasAPI greatly facilitates this process, which is presented in Claudi et al. 2020 and has extensive online documentation (https://docs.brainglobe.info/bg-atlasapi/introduction).”

3) One of the stated advantages of this software is the ability to visualize multiple types of data and data from sources that are external to the atlas generators. Such visualizations can potentially be used to reveal consistency and/or novelty across data types. However, ultimately you would want to be able to measure these differences. Including examples about how to extract and compare features across data types and then visualize those differences with brainrender would strengthen the paper.

While noting that *brainrender* is intended to be mainly a visualisation tool, we have now expanded the subsection “Example code” to include two examples of how simple analyses can be performed with *brainrender* (new Figure 6). One of the examples shows how compute the distance between the centre of mass of two injection sites, and the other shows how to extract the brain location of specific channels of a silicon probe.

4) The snapshots of code presented as figures don't add much to the manuscript. Consider highlighting how-to videos instead.

We have followed the reviewer’s suggestion and added a supplementary video illustrating how to use *brainrender*’s GUI (Video 5). Since another reviewer had the opposite opinion on the usefulness of the code snapshots, we have opted to also keep them in the manuscript.

5) As stated in earlier comments, some of the functionality is not clear, particularly for people unfamiliar or new to 3D visualization or coding in general. Even experienced developers would benefit from specs for the various input data types, for example. A little more explanation, particularly in cases where the interface is different (such as with the helper functions used to make some actors), can greatly improve the user experience.

We thank the reviewer for pointing out places in which the code and the documentation were not clear. We have now expanded the online documentation. In particular, the point about clarifying what data types are supported and how data can be loaded into *brainrender* was also raised by another reviewer, and as we mention in the reply to point 1, the documentation now includes a section dedicated to IO functionality.

We have also addressed the use of helper functions in the documentation. In brief, Streamlines and Neuron actors are most often used to visualize multiple instances of such classes at once. We thus provide helper functions to facilitate the creation of multiple Streamlines and Neuron instances. As we show in the online examples, however, these classes can also be used without the helper function, similar to the other actor classes.

6) Though the purpose of this tool is not to develop a registration algorithm for anatomical reference atlases, or to perform data analysis, we view these steps as the most difficult and necessary steps in this process. 3D rendered data visualizations are informative, but not meaningful without accurate registration to begin with and without quantitative analysis to back it up. As they point out in the introduction, other, similar tools (natverse, MagellanMapper) have both visualization and analysis capabilities.

We agree that 3D data registration is indeed a technically demanding step that is necessary before data can be visualised in *brainrender*. As we mention in our reply to point 1, the revised manuscript now discusses this explicitly and we have added additional online documentation on this topic: https://docs.brainrender.info/usage/using-your-data/registeringdata.